



**Real-time reservoir flood control operation enhanced by data assimilation**
**Jingwen Zhang[1,2], Ximing Cai[2], Xiaohui Lei[3], Pan Liu[1], Hao Wang[3]**
[1] State Key Laboratory of Water Resources and Hydropower Engineering Science, Wuhan
University, Wuhan 430072, China
[2] Department of Civil and Environmental Engineering, University of Illinois at Urbana-
Champaign, Urbana, Illinois, USA
[3] China Institute of Water Resources and Hydropower Research, Beijing 100038, China
Corresponding authors: Ximing Cai (xmcai@illinois.edu); Xiaohui Lei (lxh@iwhr.com)
**Key Points:**
• A human-machine interactive method is proposed for practical real-time reservoir flood
control operation.
• Modeling, observation, and operators' experiences are integrated for more effective
decision support for real-time reservoir operation.
• Optimization, simulation, and observation are combined for reservoir flood control via
data assimilation for long and narrow reservoirs.





**Abstract:**
Real world reservoir operations are usually not fully automatic based on computer models;
instead, reservoir operators conduct the operations based on their experiences, professional
justification, as well as modeling support for some cases due to unavoidable gap between
computer modeling and real world reservoir operation conditions. In this paper, we propose a
human-machine interactive method, namely Real-time Optimization Model Enhanced by Data
Assimilation (ROMEDA) for reservoirs which have complex storage and stage relations (e.g.
long and narrow reservoirs). The system is composed of 1) an optimization model to search for
optimal releases, 2) reservoir operators' choices based on their experiences, knowledge, and
behaviors, and 3) a reservoir storage-stage simulation and data assimilation schedule to update
the storage based on real-time reservoir stage observations. For every time period and based
on the updated storage, ROMEDA provides optimal releases as recommendations, actual
releases made by operators, as well as a warning of flood risk when the storage exceeds a
threshold level. ROMEDA does not assume that operators strictly accept the recommendations,
and storage will be updated based on actual release at each time period. Via a case study on-
channel reservoir, it is found that for both small and large flood events, ROMEDA, which
integrates the advantages of both machine and human, shows better performance on flood risk
mitigation and water use (hydropower) benefit than the case with historical operation records
(HOR) or optimization with single/multi-objective. ROMEDA is one of the first attempts of a
human-machine interactive method for online use of an optimization model for real-time
reservoir operation based on integrated modeling, observation, and operators' choice.



**Keywords:** Optimization model, human-machine interactive, data assimilation, reservoir
operation, real-time flood control
**Plain Language Summary**
Real-time reservoir flood control operation is normally controlled manually by
reservoir operators based on their experiences and justifications, rather than by computer
automatically. Computer models usually are limited in reflecting reservoir operators'
behaviors, thoughts, and priorities at particular times, resulting difficulty in direct use of the
models. In this study, we investigate how to combine machine (computer optimization model)
and human together to make the optimization model useful for real-time reservoir flood control
operation. To do this, a human-machine interactive modeling method is established to combine
computer optimization model, human's consideration, and reservoir stage observations for
actual decisions on release for real-time reservoir flood control operation. Specifically, the
optimization model provides release recommendations and a warning of flood risk; reservoir
operators determine actual release decisions based on their justification and experience based
on optimal release recommendation; however, they must deal with flood risk. To maintain the
actual reservoir storage over time, we use reservoir stage observations to update the reservoir
storage through data assimilation at each period. Via a case study reservoir, we find that real-
time reservoir flood control operation enhanced by data assimilation can reduce the flood risk
and improve water use benefit simultaneously.


## 1 Introduction

Real world reservoir operations are usually not fully automatic based on computer models; instead, reservoir operators conduct the operations based on their experiences, professional justification, and modeling support for some cases. This is because of the unavoidable gap between computer modeling and real world reservoir operation conditions (Hejazi and Cai, 2011). Especially, at present, models can hardly replace the "mental model" that is composed of experiences, knowledge, and behaviors of reservoir operators. Computer-based models for reservoir operations, especially optimization models, are usually used for "offline" analysis and providing information support for reservoir operators. Thus, it is not appropriate to assume that a model, no matter how complex it is, can be used for automatic real-time reservoir operation, although this is often the attempt of modelers.

In this paper, a human-machine interactive method is presented to support real-time reservoir operation, using reservoir flood control as an example. By this method, an optimization model for minimizing flood hazard is used for online reservoir operation via interactions with reservoir operators, as shown in Figure 1.

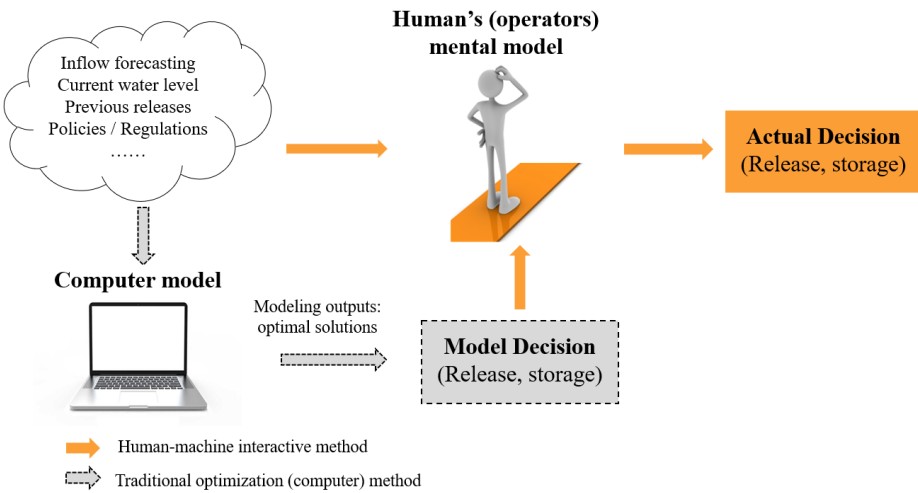

**Figure 1** The schematic of the human-machine interactive method for the online use of a computer model for reservoir operation

Many reservoir operation studies have addressed the problems of real-time optimal reservoir releases (Becker and Yeh, 1974; Chu and Yeh, 1978; Hsu and Wei, 2007). A typical real-time optimization model follows a two-stage automatic rolling-over operation scheme as shown in Figure 2: at each time period ($t$), the model determines reservoir releases at the current stage and projects releases during the periods of hydrological forecast horizon ($T$), updates the storage at the end of the period based on the release decision at the current stage, and moves forward to next time period to conduct the same modeling exercise (You and Cai, 2008; Ding et al., 2015; Draper, 2001; Draper and Lund, 2004; Zhao et al., 2012). In previous studies, such a two-stage model runs period by period and assumes that reservoir operators always follow the release provided by the optimization model at each time period (i.e., automation enabled by the optimization model).





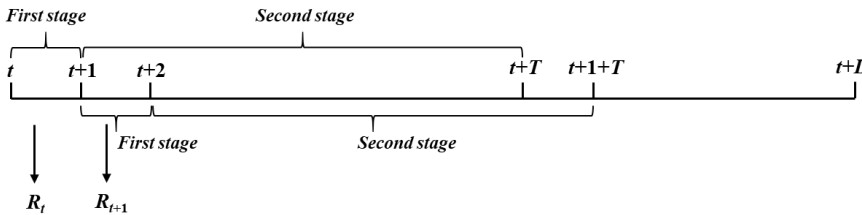


**Figure 2** Schematic of real-time two-stage rolling-over operation; $t$ represents the time step; $T$
is the forecast horizon (Zhao et al., 2019); $L$ is the remaining study period of the entire flooding
season.

Bauser et al. (2010) proposed that real-time control concept should include three parts:

real-time system simulation model, real-time observations, and optimization algorithm. Real-
time observations can be used to update the system simulation and make them close to reality.
Optimization algorithm can couple the three parts together to deliver optimal control decisions
at a rate in accordance with the response time of the real-time system (Bauser et al., 2010).
Current studies on real-time reservoir operation mostly focus on real-time system models and
optimization algorithms, aiming to explore a normative optimal solution with potential
benefits. How to use observations in real-time control system and make it useful for practical
reservoir operation remains a research challenge (Chang and Chang, 2001; Chang et al., 2005;
Dubrovin et al., 2002; Galelli et al., 2014).

The simplest method to incorporate real-time observations into real-time decision

support system is to update the model states by using the real-time observations directly as the
new states. Deng et al. (2015) used observed reservoir stages to estimate the reservoir inflow
by a simple water balance method. However, the procedure can result in inflow fluctuations
and even negative inflow values due to observation error and the uncertainty of the relationship



between reservoir storage and stage. This indicates that the direct use of real-time observations,
which ignores the model error and observation error, could lead to the error propagation. In
addition, the direct use of limited real-time observations can only update some but not all
modeling states. If the model is continuous, it is inappropriate to replace only a limited number
of model states using available observations and ignore others. Thus, the direct use of real-time
observations at limited locations or time points may end with significant errors, and combining
observations and modeling is a more effective way to simulate the continuous states of a
process (Crow and Loon, 2006; Huang et al., 2002; Trenberth et al., 2008).

Real-time observations are usually incorporated via more sophisticated data

assimilation techniques to improve dynamic modeling, as demonstrated by numerous modeling
efforts in ocean modeling (Evensen, 1994; Carton and Giese, 2008; Oke et al., 2005), weather
forecasting (Kanamitsu, 1989; Houtekamer and Mitchell, 1998; Barker et al., 2004),
hydrological modeling (Xie and Zhang, 2010; Reichle et al., 2008; Wang and Cai, 2008), etc.
Data assimilation has been also applied to the water resources system modeling for more
efficient operation (Bauser et al., 2010; Munier et al., 2015). Bauser et al. (2010) used an
optimal real-time control approach with data assimilation to manage the urban groundwater
well fields to reduce diffuse pollution in the Hardhof field of Zurich, Switzerland. Ensemble
Kalman Filter (EnKF) was applied to incorporate 87 online groundwater head observations
into a three-dimensional finite element subsurface flow model for real-time allocation of
artificial recharge. Munier et al. (2015) applied data assimilation for operational water
management on the upper Niger River Basin. The virtual Surface Water and Ocean
Topography (SWOT) observations of reservoir and river levels with a repeat cycle of 21 days



were assimilated to initialize a model predictive control algorithm for optimal reservoir
operation. These studies showed that water resources management supported by the
assimilation of real-time observations outperformed the optimization models without the online
data support.

This study utilizes data assimilation to connect reservoir optimization-simulation

models and observations resulting from actual reservoir releases decisions. Many previous
studies on real-time reservoir operation optimization used a simple lumped water balance
model to represent the reservoir dynamics (Galelli et al., 2014), or simply use the observed
stages and the storage-stage relationship to calculate the reservoir storage. In this study we
demonstrate that an unsteady flow routing simulation model is needed for reservoirs that are a
long and narrow channel, for which it is not accurate enough to use a static storage-stage
relationship to simulate the reservoir storage; while it is also impossible to measure the storage
directly because the reservoir surface is not flat. This special case, which exists for many large
and long reservoirs around the world, solicits the use of the data assimilation technique to
enhance the accuracy of the unsteady flow routing model using observed stages at different
sections along the reservoir channel to update model states and also control model and
observation errors.

The primary goal of the present paper is to combine the traditional optimization model

(i.e. computer model) and human's consideration together for real use of an optimization model
on real-time reservoir flood control. This paper is to address the following three questions: (1)
How can the computer model and human's consideration be combined for online real-time
reservoir operation? (2) What is the performance of the combined method compared to the



actual operation or the result of the optimization model? (3) What is the impact of observations
on the real-time reservoir flood control? To answer these questions, we propose the Real-time
Optimization Model Enhanced by Data Assimilation (ROMEDA) via a human-machine
interactive method with the assimilation of real-time observations. Observed data, reservoir
operators' choices, and computer models will be coupled in the ROMEDA. In the rest of this
paper, we start with an overview of the two methods (ROMEDA method and OPT method)
and detailed introduction of ROMEDA. Then, an example of an on-channel reservoir for flood
control is used to demonstrate ROMEDA. Finally, the discussion on performances and
characteristics of ROMEDA is compared to those of the optimization model and historical
operation records (HOR).
**2 Methodology**

2.1 Overview

For the real-time rolling-over reservoir operation, the OPT method, i.e. computer

models, determines the optimal releases with the current storage and forecasted inflow at every
time period ($t$) (Figure 3). The optimal release is automatically assumed as the actual release
and taken as input into an on-channel reservoir system simulation model to calculate the stages
at all cross sections of the channel upstream of the dam, based on which, the simulated reservoir
storage is then determined as the initial storage for the next time period. The on-channel
reservoir system model is a one-dimensional (1-D) hydrodynamic model during flood events
in the Preissmann scheme (Preissmann, 1961).





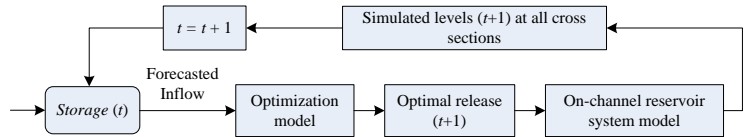


**Figure 3** Scheme diagram of the OPTimization (OPT) method
The ROMEDA method is illustrated in Figure 4. The real-time optimization model can
provide the optimal releases at each time period of the entire study period. The reservoir
operators can choose to take the result provided by the optimization model or any decision
based on their own priority for the current time period ($t$). At the end of period $t$, the reservoir
storage will be updated based on what the operators' choice of reservoir releases and the
optimization model will decide the optimal releases for period $t+1$ based on the updated
reservoir storage (i.e., the state variable) and the inflow forecast for the rest of the study period.
This procedure will be continued till the end of flooding season. The essential difference
between this method and the direct use of OPT is the online incorporation of 1) reservoir
operators' choices based on their experiences, knowledge and behaviors to determine actual
reservoir releases; 2) the real-time observation of stages along the channel upstream of the dam
to update the reservoir storage so as to provide the optimal release based on actual storage.
Actually, the operators can choose when to adopt the modeling results themselves. This is
because reservoir operators' considerations vary by person and by reservoir. In this paper, as
an illustration example, we set that reservoir operators adopt modeling results when the storage
is over the maximum storage required for leaving space for coming storms. This is only one of
the possible ways of the operators may choose via the human-machine interactive method.





A data assimilation method is used to assimilate the observations at some channel
sections to the on-channel reservoir system simulation model, taking account of both the model
error and observation error, to update the stages at all cross sections. In this way, the stage
resulting from the actual decision at time period $t$ is observed and assimilated to simulate the
storage at time period $t+1$, which is taken as real-time input for the optimization model. Thus,
compared to the OPT method, ROMEDA provides release decision recommendations for
reservoir operators period by period and does not assume all the recommendations will be
adopted by the reservoir operators. In addition, an advanced data assimilation algorithm,
Constrained Ensemble Kalman Filter with accept/reject method (Wang et al., 2009), to be used
in the ROMEDA, will handle the impact of both model and observation errors, as detailed later.

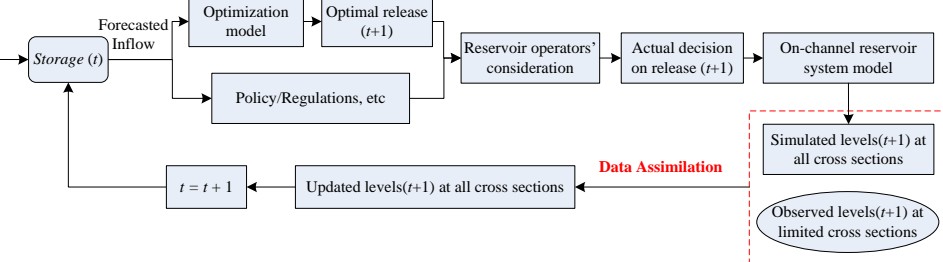


**Figure 4** Scheme diagram of the Real-time Optimization Model Enhanced by Data
Assimilation (ROMEDA) method
ROMEDA is similar to Model Predictive Control (MPC) (Garcia et al., 1989; Camacho
and Alba, 2013; Macian-Sorribes and Pulido-Velazquez, 2019) and other real time control
approaches, such as on-line adaptive control (Soncini-Sessa et al., 2007), open-loop and closed-
loop control (Soncini-Sessa et al., 2007; Gerdts, 2012) with respect to more effective use of
computer-based models and observed data. MPC conducts rolling-horizon optimization based
on state observations and input forecasts. Essentially, MPC targets a computer-based automatic
operation program; while ROMEDA follows the human-machine interactive method. Both
ROMEDA and MPC update the model states using observations at each time step. However,
MPC methods handle predictive environmental disturbance, such as weather forecast
uncertainty (Ficchì et al., 2015; Raso et al., 2014; Maestre et al., 2012); while ROMEDA
integrates operators' choices with the solutions from a computer model. Particularly, MPC
methods usually use the observed data directly; while ROMEDA assimilates observed stages
via a data assimilation technique to update the simulation of reservoir storage. Thus, ROMEDA
couples optimization, simulation, data assimilation, and human choices; the method is tested
with real-time reservoir operation for flood control in this paper.
2.2 Real-time modeling of an on-channel reservoir system for flood control
2.2.1 1-D hydrodynamic model
The 1-D unsteady flow routing in on-channel reservoir system can be described by the
Saint-Venant equations, including the continuity equation and the momentum equation, as
follows:

$$\frac{\partial A}{\partial t} + \frac{\partial Q}{\partial x} - q = 0 \tag{1}$$

$$\frac{1}{A}\frac{\partial Q}{\partial t} + \frac{1}{A}\frac{\partial}{\partial x}\left(\frac{Q^2}{A}\right) + g\frac{\partial z}{\partial x} - g\frac{n^2 Q|Q|}{AR^{4/3}} = 0 \tag{2}$$

where $A$ is active flow area, i.e. the proportion of the total cross-sectional area with flow; $Q$ is
the streamflow; $q$ is the lateral inflow/outflow per unit length, including the runoff generated
along the river channel; $x$ and $t$ are the independent variables of space and time, respectively;
$g$ is the acceleration due to gravity; $z$ is the depth of flow; $n$ is the roughness coefficient; and $R$
is the hydraulic radius, $R = \dfrac{A}{\chi}$; and $\chi$ is the wetted perimeter. The Preissmann implicit four-
point finite difference scheme, a widely used numerical method, is used to solve the 1D
hydrodynamic model (Preissmann, 1961; Castellarin et al., 2009). The streamflow and stages
at all cross sections of the channel upstream of the dam at next time period $t$+1 can be
determined with the boundary conditions (streamflow at the first and last cross sections, i.e.
inflow and release) at period $t$+1 and the streamflow and stages at all cross sections at period
$t$. The water storage between two adjacent cross sections can be calculated as the volume of
prismoid. Thus, the reservoir storage can be determined by accumulating the storage between
all adjacent cross sections. The details of the Preissmann scheme should be referred to
Appendix A.
2.2.2 Real-time reservoir optimization model
Flood control is the primary objective during the flooding season, and the tradeoff
between the upstream and downstream flooding damage is a longstanding challenge for
reservoir operation. To account for the tradeoff, the real-time reservoir deterministic
optimization model with a short forecast horizon can be set up with a single objective to
minimize the maximum reservoir storage during the forecast horizon (Eq. 3) subject to a
constraint on the maximum release for downstream. However, the reservoir operators'
consideration could go beyond the sole flood control objective even during the flooding season,
and consider to minimize hydropower generation loss during and after the flood control period.
Thus the optimization can also be set up with multi-objectives, i.e., one for flood control and
the other for maximizing hydropower generation (Eq. 4).





$$\min \ OBJ^* \Leftrightarrow \min \ \left[ \max S(t) \right] \tag{3}$$

$$\begin{cases} \min \ OBJ_1^* \Leftrightarrow \min \ \left[ \max S(t) \right] \\ \max \ OBJ_2^* \Leftrightarrow \max \ \left[ \sum_{t}^{t+T} P_t \right] \end{cases} \tag{4}$$

where $\max S(t)$ is the maximum reservoir storage during the forecast horizon ($T$); and $P_t$ is
the hydropower generation during time period $t$.

The major constraints include the lower and/or upper bounds for reservoir release,

stages at all cross sections, storage, power generation output, and the largest incremental
release between consecutive time periods:

$$R(t) \le R_{\max} \tag{5}$$

$$Z_{\min}^j \le Z^j(t) \le Z_{\max}^j \tag{6}$$

$$S_{\min} \le S(t) \le S_{\max} \tag{7}$$

$$PL(t) \le P(t) \le PU(t) \tag{8}$$

$$\left| R(t) - R(t+1) \right| \le \Delta R \tag{9}$$

where $R(t)$ and $R(t+1)$ are the reservoir releases during time period $t$ and $t+1$, respectively;
$R_{\max}$ is the maximum allowed release during the flood event; $Z^j(t)$ is the stage at cross
section $j$ for the on-channel reservoir at time period $t$; $Z_{\min}^j$ and $Z_{\max}^j$ are the minimum and
maximum allowed stage at cross section $j$ for the on-channel reservoir; $S_{\min}$ and $S_{\max}$ are the
minimum and maximum allowed storage for the on-channel reservoir; $PL(t)$ and $PU(t)$ are
the minimum and maximum hydropower generation output limits for the on-channel reservoir
during time period $t$; $\Delta R$ is the allowed maximum incremental release over consecutive
periods.



The forecast horizon of the real-time reservoir flood control model is 3 days with a 1-

hour time step. At every time period, the 72 hourly releases during the forecast horizon are the
decision variables. Stochastic global optimization algorithms, Dynamically Dimensioned
Search algorithm (DDS) (Tolson and Shoemaker, 2007) and Pareto archived dynamically
dimensioned search algorithm (PADDS) (Jahanpour et al., 2018), are applied to find the
optimal releases at each time period for a single objective or multi-objective optimization
model (OPT-S and OPT-M). The maximum number of function evaluations with the above
steps is set to 1,000 for every time period by DDS and PADDS. DDS and PADDS can converge
to good solutions rapidly and avoid the poor local optima.

2.3. Data assimilation

Data assimilation techniques can effectively estimate the states of a complex system

with the observations. Ensemble Kalman Filter (EnKF), a sequential data assimilation scheme,
has been widely used in hydrological modeling (Botto et al., 2018; Liu and Gupta, 2007;
Moradkhani et al., 2005; Feng et al., 2017). Two processes, i.e. forecasting and updating
processes, constitute the EnKF framework, described by:

$$Z_{t+1|t}^k = f\left(Z_{t|t}^k, n\right) + \omega_t^k, \omega_t^k \sim N\left(0, W_t\right) \tag{10}$$

$$Z_{t+1|t+1}^k = Z_{t+1|t}^k + K_{t+1}\left[Z_{t+1}^{obs,k} - h\left(Z_{t+1|t}^k\right)\right] \tag{11}$$

$$Z_{t+1}^{obs,k} = Z_{t+1}^{obs} + \upsilon_{t+1}^k, \upsilon_{t+1}^k \sim N\left(0, V_{t+1}\right) \tag{12}$$

where $Z_{t|t}^k, Z_{t+1|t+1}^k$ are the $k^{th}$ updated ensemble member of the stage vector at time period $t$
and $t$+1; $Z_{t+1|t}^k$ is the $k^{th}$ forecasted ensemble member of stage vector at time period $t$+1; $n$ is
the system parameter, i.e. roughness coefficient (see Appendix B); $f$ represents the system





model; $Z_{t+1}^{obs,k}$ is the perturbed observed stage of selected cross sections of $k$th ensemble
member at time period $t+1$, obtained by adding Gaussian observation error $\upsilon_{t+1}^{k}$ to the
observation $Z_{t+1}^{obs}$; $h$ is the observation function, i.e. selecting the forecasted stage at selected
cross sections, corresponding to the observations; $\omega_{t}^{k}$ and $\upsilon_{t+1}^{k}$ are the system model error and
Gaussian observation error, which are assumed to follow Gaussian distribution with zero mean
and specified diagonal covariance matrix $W_{t}$ and $V_{t}$. The standard derivation of model state
errors and observation errors are set as 0.5 and 0.01, respectively; and $K_{t+1}$ is the Kalman gain
matrix.

As the forecasted states and updated states may violate the states constraints,

Constrained Ensemble Kalman Filter (CEnKF) is proposed to deal with the violations (Pan and
Wood, 2006; Wang et al., 2009). Because of its computational efficiency and modeling
accuracy, CEnKF with the Accept/Reject Method is used in this paper (Wang et al., 2009). All
constraints in the forecasted and updated states are checked in the forecasting and updating
processes, respectively. A threshold of maximum number of rejections, 500, is set for each
ensemble member at every period to limit the computational burden. If the forecasted/updated
states still violate the constraints when the number of rejections reaches the threshold, the loop
stops, and the states would be set to the boundary directly. The details of the data assimilation
procedures can be referred in Wang et al. (2009).



**3 Case study**

3.1 An on-channel reservoir system

An on-channel reservoir for flood control from China is selected to test the proposed

ROMEDA method. The reservoir receives inflow from a drainage area of 56,000 km$^2$. The
flood control capacity of the reservoir is 22.2 km$^3$. The channel upstream of the dam has a
length of 658 km and the average width of the channel is 1.1 km. Figure 5 shows some selected
sections of the channel. Due to the geometry and topological characteristics (i.e., a long and
narrow channel upstream of the dam), the flood wave propagation requires about 24-36 hours
from the upstream tail of the reservoir to the dam location. The surface of the on-channel
reservoir featured by a significant slope cannot be treated as flat during the flooding season.
Thus, it is not appropriate to simulate the reservoir flood routing by static storage-stage
relationship assuming a flat surface. A 1-D unsteady flow routing model is used to simulate
flood routing in the on-channel reservoir, by which the dynamic reservoir storage is calculated
using a numerical method. Figure 6 shows the longitudinal profile of the bottom elevation of
296 cross sections in the upstream channel of the dam, as well as the reservoir surface. Stage
observations can be obtained from 11 sections as shown in Figures 5 and 6. The characteristic
parameters of the on-channel reservoir are listed in Table 1. It should be noted that hydropower
generation is one of the major functions of the on-channel reservoir, with an installation
capacity of 22,400 MW.



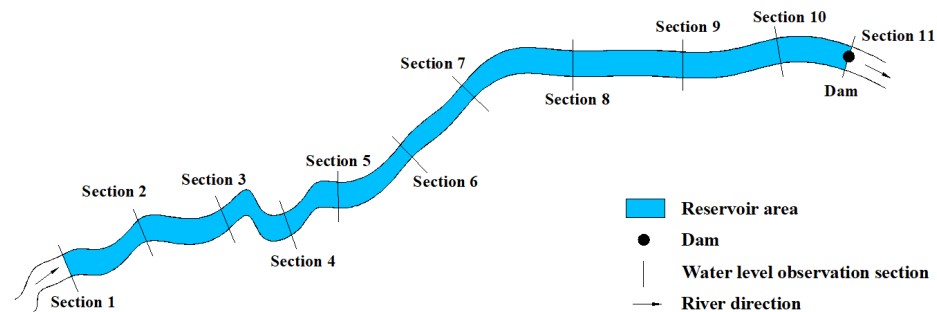


**Figure 5** Schematic diagram of the river and on-channel reservoir


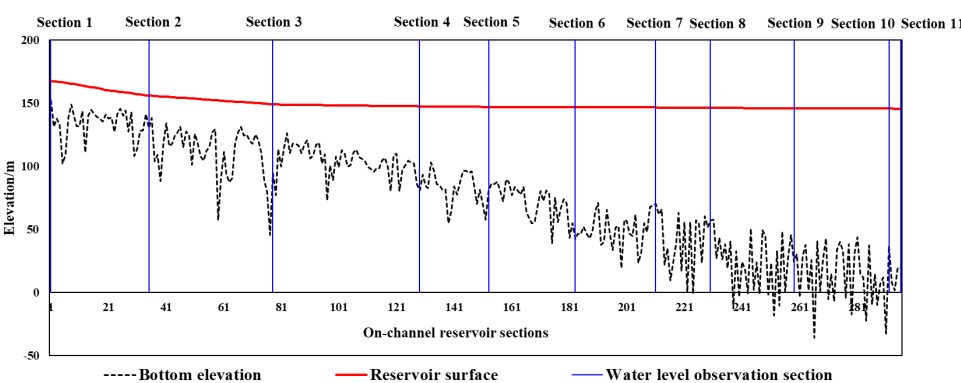


**Figure 6** Longitudinal profile of the on-channel reservoir


**Table 1** Characteristic parameters of the on-channel reservoir

| Flood limited stage (m) | Normal pool stage (m) | Crest elevation (m) | Flood protection storage (km³) | Total reservoir storage (km³) |
|---|---|---|---|---|
| 145 | 175 | 185 | 22.15 | 39.3 |

3.2 Data

Two historical flood events with different magnitudes (small and large) with a time step

of 1 hour are selected for case studies. Since the forecast horizon of the real-time reservoir
flood control model is set to be 3 days with a 1-hour time step, the forecast uncertainty is




relatively low, historical inflows are used as perfect inflow forecast without uncertainty. The
stage observations at the 11 observation sections (Figure 6) are provided with the same time
step (1 hour) during both the small and large flood events. The real-time stage observations
provided to ROMEDA for the case study use the historical stage records resulting from actual
releases during the periods when reservoir operators do not adopt the results from OPT;
otherwise "virtual stage observations" resulting from the OPT suggested releases (adopted by
the operators) are used. The maximum allowed release of the on-channel reservoir during the
flooding season varies depending on the flood magnitudes. In this paper, the maximum allowed
releases for small and large flood events are set to 29,800 $m^3$/s and 43,300 $m^3$/s, respectively.
The storage threshold for flood risk is 22.8 $km^3$.
**4 Results**
The performance of OPT-S, OPT-M, ROMEDA, and the historical operation records
(HOR) on flood risk and water use benefit are compared in the section. During the flooding
season, the reservoir operators are required to follow the optimal operation aiming to reduce
the flood risk; meanwhile they may have other considerations, such as considering the
maximum allowed flow specified by the hydropower installation capacity to reduce spill. Due
to complex and variable human's considerations, we test a particular case for simplicity but
without losing generality, i.e., reservoir operators do not necessarily follow the recommended
releases when the current reservoir storage is lower than the storage threshold for the case
reservoir (22.8 $km^3$), but will do it when the storage exceeds the storage threshold.





4.1 Operation processes
The modeling results from the three methods (OPT-S, OPT-M, and ROMEDA), along
with the historical operation records (HOR), are compared in Figure 7 for a small flood event
and Figure 8 for a large flood event. The maximum releases under all these cases reach to the
maximum allowed release (29,800 m³/s for a small flood event and 43,300 m³/s for a large
flood event). OPT-S, driven by the objective of minimizing the peak storage during a flood
event, releases more water to reserve a large flood control storage for future possible flood
events. Under OPT-M, the releases are slightly smaller than that of OPT-S before the flood
peak, which is driven by the objective of maximizing the hydropower generation. As shown in
Figure 7, before the arrival of the first flood peak (during the first 150 time periods in Figure
7a), the releases of OPT-S and OPT-M are larger than those of HOR (but smaller than the
maximum allowed release, 29,800 m³/s). Given the forecast of the first and second coming
inflow peaks, the OPT-S and OPT-M releases increase sharply and reach to the highest allowed
level during period 150-180. After the arrival of two flood peaks, the OPT-S and OPT-M
releases, though lower than the maximum allowed release, can make the reservoir storage lower
than the threshold level during all modeling periods.
As stated above, the reservoir operators' consideration could go beyond the sole flood
control objective, as avoiding hydropower generation loss during and after the flood control
period is also of concern. Consequently, they may release less water than OPT-S and OPT-M
prescribes to reserve a larger water storage that is beneficial for hydropower generation. As
shown in Figure 7b, rather than a sharp increase (500 m³/s per hour) under OPT-S and OPT-
M, the HOR releases during periods 120 to 230, only gradually increase (170 m³/s per hour),





which eventually ends with a reservoir storage that exceeds the storage threshold. During the
flood peak period, the HOR release is high but smaller than the maximum allowed release
(while the OPT-S and OPT-M releases approximately equal the maximum allowed release),
which makes the storage under HOR continuously remain above the storage threshold (Figure
7c). After the peak period (period 230 and further), the HOR releases approximately reach the
maximum allowed release to reduce the flood risk since the reservoir storage is still above the
threshold. In summary, the real-world situation (HOR) could be complicated by several
conditions: first, they might take a certain level of risk of flooding for the benefits of
hydropower (i.e., dealing the tradeoff). Second, the operation of the reservoir does not exactly
follow the flood control requirements (i.e., the storage is over the threshold during some
periods). Third, the actual releases might also be affected by the requirement of the maximum
allowed releases to downstream.

As assumed, the releases of ROMEDA are basically the same as those of HOR to

maintain water use benefits, when the reservoir storage does not exceed the storage threshold.
The reservoir storage of ROMEDA is updated with the assimilation of real-time reservoir
observed stages, which can mitigate the model error from the 1-D hydrodynamic model. When
the reservoir storage, updated via ROMEDA, exceeds the storage threshold, the reservoir
operators follow the recommended releases from the optimization model in order to reduce the
flood risk. This ends with a large increase of the releases compared to those before the storage
reaches the threshold since the recommended release is close to the maximum allowed release.
By doing that, the reservoir storage of ROMEDA is decreased to the threshold during the flood
peak. After the flood peaks, when the storage is below the threshold, the releases of ROMEDA
come back to the HOR releases (after period 230). Due to the impact of peak-clipping
conducted during the flood peak periods, the reservoir storage of ROMEDA is lower than that
of HOR with the same reservoir release after two flood peaks (Figure 7c). Overall, ROMEDA
reduces the flood risk with large releases from the optimization model and meanwhile increases
the hydropower generation from the HOR (see more discussion in the following).

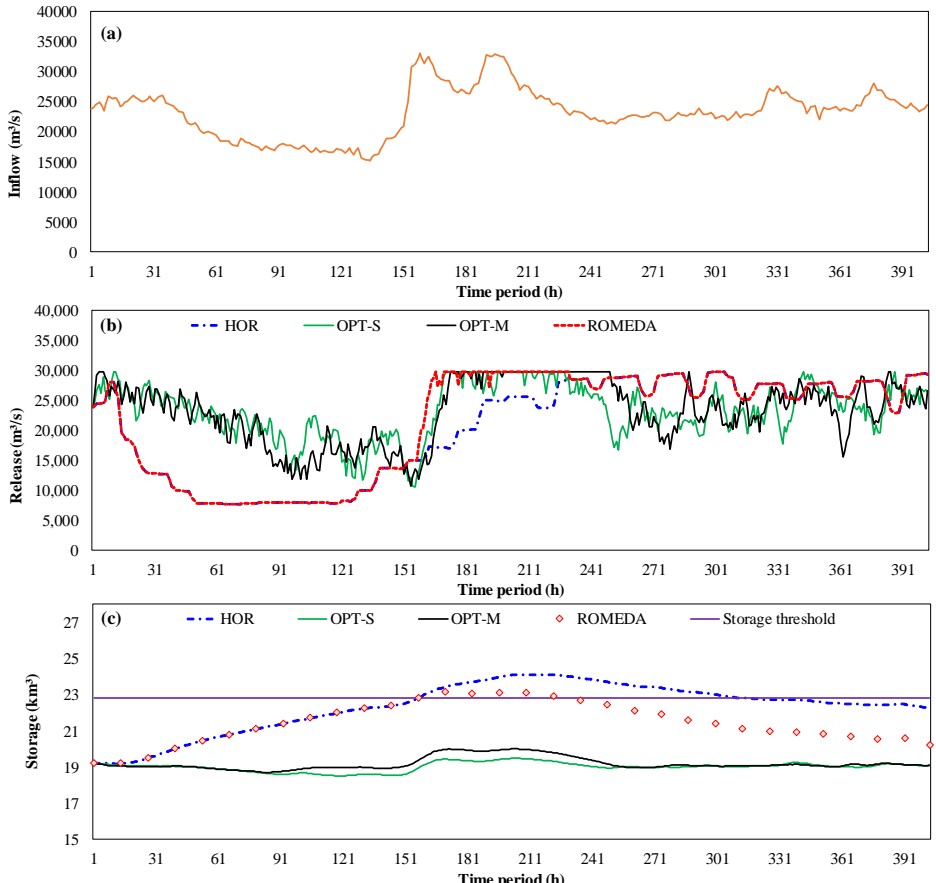


**Figure 7** Reservoir operation results of HOR, OPT-S, OPT-M, and ROMEDA four cases for a

small flood event (a) inflow; (b) release, ROMEDA denotes the adopted release in ROMEDA
case; and (c) storage, ROMEDA denotes the updated storage in ROMEDA case





Reservoir operation results of the four cases (HOR, OPT-S, OPT-M, and ROMEDA)

for a large flood event are compared in Figure 8. There are three flood peaks in this flood event.
During the first peak, OPT-S releases more water than HOR to reserve the flood control storage
for possible flood peak in future, resulting in smaller reservoir storage under OPT-S than HOR;
while OPT-M releases less water to increase the reservoir storage for larger hydropower
generation. HOR releases increase to the maximum allowed flow specified by the hydropower
installation capacity before the first flood peak, aiming to reduce spill during the flood peak
periods. It seems that the HOR releases correspond to the inflow variability and the reservoir
storage is within the storage threshold though it is higher than those of OPT-S and OPT-M.
After the first flood peak, the HOR releases keep close to the maximum allowed release; while
the OPT-S and OPT-M releases exhibit variations due to the sensitivity to the flood forecast
(noted that the variations can also be caused by the uncertainty of the optimal solution from the
DDS and PADDS algorithms). It is found that the reservoir storage of OPT-S and OPT-M
exceed that of HOR during the second flood peak. Meanwhile, for both HOR and OPT-S, the
releases are close to the maximum allowed release (43,300 $m^3$/s) and the reservoir storage
exceeds the threshold, but OPT-M releases are smaller than the maximum allowed release due
to objective of hydropower generation. After the second flood peak, the storage of both OPT-
S, OPT-M, and HOR still remains above the storage threshold for some periods, but HOR
gradually reduces the releases to the maximum allowed flow designed for the installation
capacity of the hydropower station; while OPT-S and OPT-M keeps the releases at the
maximum level for longer periods. Thus, OPT-S and OPT-M have less periods with its storage
above the threshold and accumulated value of flood risk than HOR (see more discussion in





Section 4.2). Overall, compared to a small flood event, HOR, OPT-S, and OPT-M respond to
inflow variability closely during a large flood event, and both end with some periods with
storage over the threshold level. OPT-M has the largest value of maximum reservoir storage.
Meanwhile the HOR releases and storage still imply some considerations beyond flood control.

Before period 310, the reservoir storage of ROMEDA is below the storage threshold,

and the ROMEDA releases are the same as those of HOR. After that ROMEDA takes the
recommended releases from the optimization model during the period from the second flood
peak to the end of the third flood peak, during which the ROMEDA storage is over the threshold
level too but having a smaller number of periods with its storage above the threshold than HOR
and OPT-M. After the three flood peaks, the ROMEDA releases come back to the releases of
HOR, and the ROMEDA storage is larger than that of OPT-S and smaller than that of HOR,
which shows a balance of flood control and hydropower generation.

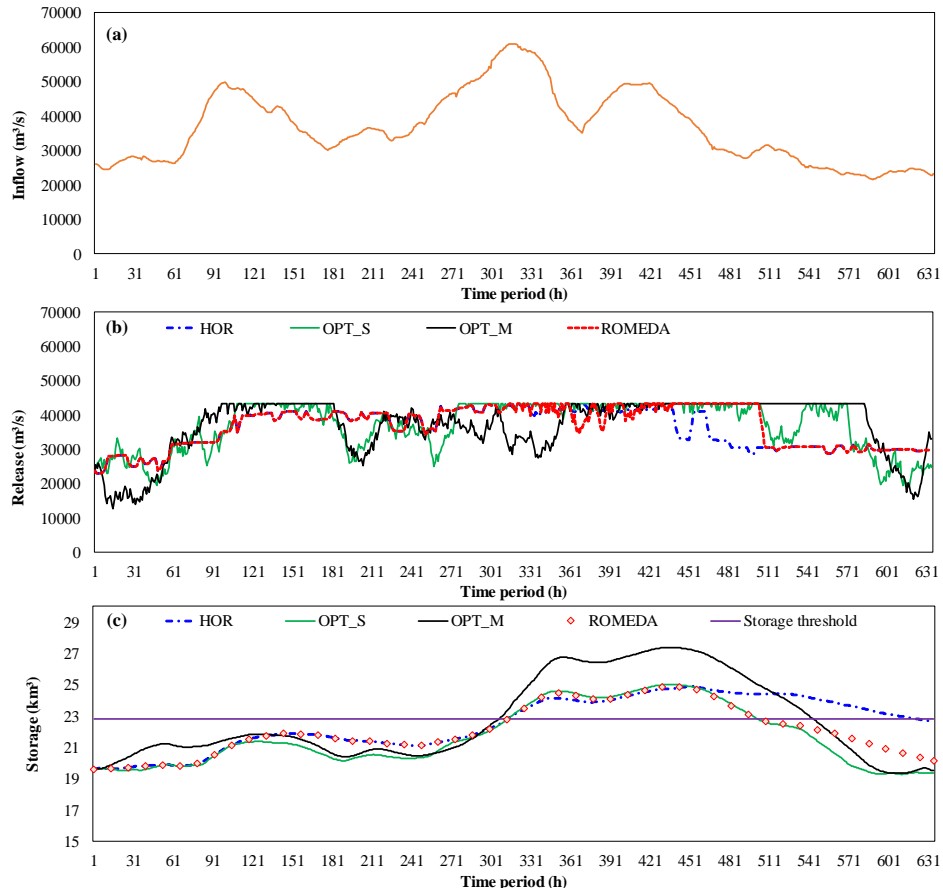


**Figure 8** Reservoir operation results of HOR, OPT-S, OPT-M, and ROMEDA four cases for a

large flood event (a) inflow; (b) release, ROMEDA denotes the adopted release in ROMEDA

case; and (c) storage, ROMEDA denotes the updated storage in ROMEDA case

As shown in Figure 7b, under the small flood event, the HOR releases increase from

the minimum to the maximum taking 110 periods (from period 120 to 230), accounting for

28% of the total periods; the maximum release remains 170 time periods (from period 230 to

400), 43% of the total periods. However, under the large flood event, HOR only takes 70 time

periods (11% of the total flood periods) to increase from the minimum to the maximum release





as shown in Figure 8b; the maximum release remains 310 time periods (from period 120 to
430), almost half of the total periods. These results indicate that reservoir operators in HOR
behave differently when they deal with small and large flood events. It seems that the reservoir
operators have aggressive behaviors toward the tradeoff between flood control and hydropower
generation during a small flood event, while they are more conservative during a large flood
event by taking quicker and stronger measures for peak-clipping.

4.2 Flood risk vs. water use benefit

The performance of HOR, OPT-S, OPT-M, and ROMEDA four cases are further

compared in terms of flood risk and water use benefit. Flood risk is triggered when the reservoir
storage exceeds the threshold level. Besides the maximum reservoir storage and maximum
stage in front of the dam, we choose the number of periods with storage over the threshold and
the accumulated value of flood risk over time as two indicators of flood risk. The accumulated
value of flood risk is calculated as the sum of reservoir storage amount exceeding the threshold
level during the entire flood event. Figure 9 shows the comparison of the indicators of HOR,
OPT-S, OPT-M, and ROMEDA under a small and a large flood event.




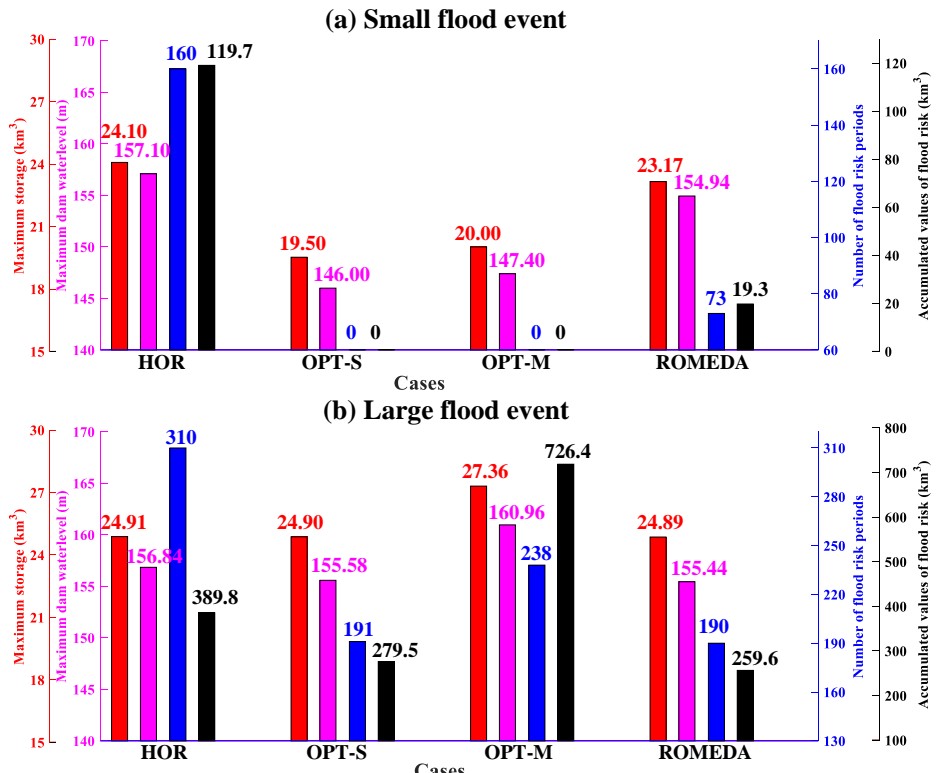

**Figure 9** Four indicators of flood risk (maximum storage, maximum stage in front of the dam, number of flood risk periods, and accumulated values) among HOR, OPT-S, OPT-M, and ROMEDA four cases for (a) a small flood event, and (b) a large flood event

For the small flood event, OPT-S and OPT-M have the lowest maximum reservoir storage and maximum stage in front of the dam, and also have zero risk indicators of flood risk periods and accumulated values; ROMEDA has lower values of four indicators than HOR. In particular, the risk periods and the accumulated risk (i.e., the sum of reservoir storage amount exceeding the threshold level during the entire flood event) are largely reduced under ROMEDA. As can be seen in Figure 7c, during some periods, the reservoir storage of ROMEDA exceeds but is close to the threshold level.

For the large flood event, OPT-M has the largest maximum reservoir storage (27.36
km$^3$), maximum stage in front of the dam (160.96 m), and the accumulated value of flood risk
(726.4 km$^3$) due to the maximization of hydropower generation in the multi-objective
optimization context. HOR has the largest number of flood risk periods (310 periods). The
performance of OPT-S and ROMEDA are close, but ROMEDA has the lowest values of four
indicators among the four cases , indicating that the conservative behaviors of reservoir
operators as reflected in HOR have an important influence on flood risk reduction with a large
flood event. Thus, the proposed ROMEDA performs well in terms of flood risk reduction by
combining the optimization model results and the experiences of reservoir operators.

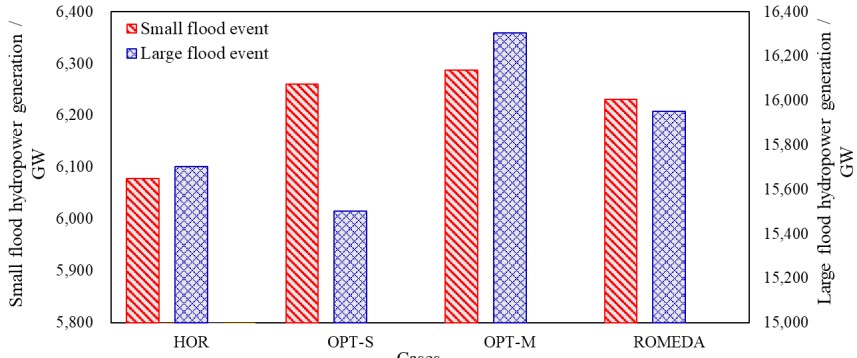


**Figure 10** Hydropower generation comparison among HOR, OPT-S, OPT-M, and ROMEDA
for a small and a large flood event.
To further compare the four cases with respect to the reservoir operation purpose
beyond flood control, Figure 10 displays the hydropower generation during a small and a large
flood event given that in the case study reservoir, hydropower is the major objective for the
reservoir operators subject to flood control requirements. As there is a magnitude difference of
inflow between a small and a large flood event, the hydropower generation of the four cases



under the small and the large flood event is compared with different scales as shown in Figure
10. As expected, OPT-M has the largest hydropower generation among the four cases given its
multi-objective of flood control and hydropower generation with a small and a large flood
event. OPT-S results in the lowest hydropower generation among the four cases given its sole
objective of flood control in a large flood event. The performance of ROMEDA is between
OPT-M and HOR in a small and a large flood event, i.e. the hydropower generation of
ROMEDA is lower than that of OPT-M but higher than that of HOR. The amount of
hydropower generation depends on release and the hydraulic head (which usually has a
consistent relationship with reservoir storage). As can be seen from Figure 7, for the small
flood event, HOR takes low releases during the first and second flood peak in order to store
water to build a high hydraulic head. While by following the recommendation from the
optimization model, ROMEDA take higher releases than HOR. In general, the situation in a
large flood event is the same as that in a small flood event (Figure 8). Eventually the associated
levels of release and head under ROMEDA results in higher hydropower generation than HOR.
Therefore, compared to ROMEDA, HOR results in a low water use profit and high flood risk
under a small and a large flood event. This implies that the practices of the reservoir operators
can be improved by a model which may respond more closely to the current state of the
reservoir and forecasts of future inflow.



## 5 Discussions

The effects of assimilating real-time observations for modeling error correction, the form of the objective function, and inflow forecast uncertainty on the performance of ROMEDA are discussed in this section.

5. 1 Effects of real-time observations

ROMEDA, a human-machine interactive method, utilizes data assimilation to connect reservoir optimization-simulation models and observations resulting from actual reservoir releases. Data assimilation of real-time observed stages at different sections along the reservoir channel can reduce model errors and enhance the accuracy of the unsteady flow routing model.

Figure 11 shows the effectiveness of the assimilation of stage observations on eliminating the model errors on the reservoir storage under HOR by two cases with and without data assimilation under the small and large flood events. Under one case, the storage is calculated using the same inputs of historical inflows and releases using the numerical Preissmann scheme directly (OPT also takes this method); under the other case, the storage is simulated by the scheme along with the assimilation of stage observations. As shown in Figure 11a, the storage simulated without data assimilation is larger than that with data assimilation, indicating that for a small flood event, the on-channel reservoir system simulation model may overestimate the reservoir storage. The overestimated storage may mislead reservoir operators into paying additional unnecessary attention to flood control, which may result in unnecessary loss of hydropower generation. However, the opposite result can be seen for a large flood event in Figure 11b, i.e., the storage simulated without data assimilation is underestimated, which

may mislead reservoir operators into underestimating flood risk. This further confirms the
advantage of the data assimilation in ROMEDA method by mitigating modeling errors and
enhancing the effectiveness of the modeling work.

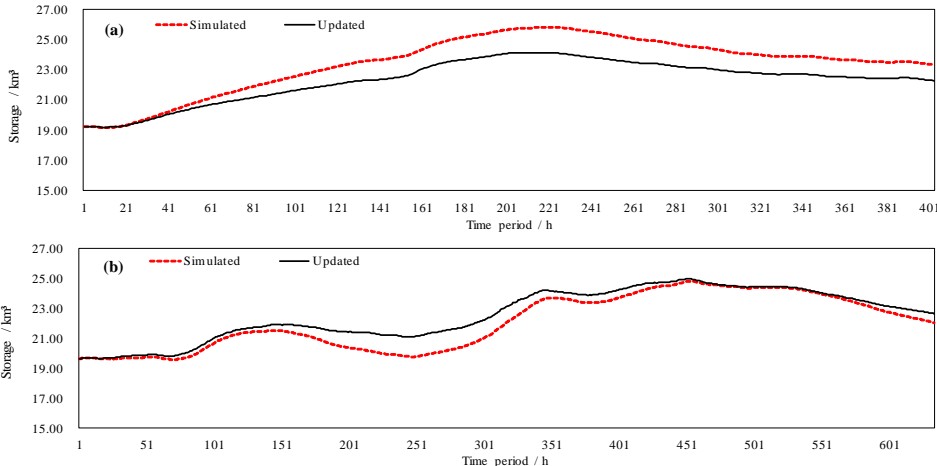


**Figure 11** The historical storage difference resulted from the model error for (a) small and (b)
large flood events
5. 2 On the form of the objective function of OPT
An optimization model can always be improved to make it closer to the "idea one".
However, a realistic way to use a model is to combine the model result with operators' choices
based on their experiences, knowledge, and behaviors. In this study, flood control is the
primary objective during flooding season, particularly the flood peak period. For the case study
reservoir to test the proposed method, reservoir operators also try to avoid large reduction of
hydro-energy generation. Thus, we set up a single-objective optimization model (OPT-S)
considering flood control and a multi-objective optimization model (OPT-M) considering both
flood control and hydropower generation. Based on the comparison of flood risk and



hydropower generation among the four cases (HOR, OPT-S, OPT-M, and ROMEDA) in Figure
9 and 10, we found that OPT-S and OPT-M can both achieve better performance of flood risk
and hydropower generation than HOR and ROMEDA in a small flood event (Figure 9a and
10); while OPT-S has a smaller value of hydropower generation and OPT-M has larger flood
risk indicators than HOR and ROMEDA in a large flood event (Figure 9b and 10). This
indicates that different objective combinations have different performances for flood events
with different magnitudes. However, ROMEDA, which incorporates the
knowledge/experience of reservoir operators and the outputs of the optimization model, can
achieve a good performance of flood risk reduction and hydropower generation in both a small
and a large flood event (Figure 9 and 10). This indicates that ROMEDA makes more effective
use of the reservoir operators' experience and the optimization model.
5.3 On forecast uncertainty
Weather forecast uncertainty is vital to real-time reservoir flood control. The forecast
uncertainty especially with long lead time (Zhao and Zhao, 2014) with climate/weather
variables such as precipitation (Saavedra Valeriano et al., 2010) or hydrologic variable such as
reservoir inflows (Maurer and Lettenmaier, 2004) all have significant impact on real-time
reservoir operation. In this paper, historical inflow is used as "perfect inflow forecast" to test
the proposed method, underlying an assumption that the uncertainty level of the forecasts with
relatively short heading time horizon (72-hour) is low. While the focus of this paper is to
demonstrate the human-machine interactive method, it can be extended to account forecast
uncertainty, for example, by adopting Model Predictive Control (MPC) (Galelli et al., 2014;
Ficchì et al., 2015).





**6 Conclusions**

Reservoir operation models, especially optimization models are usually suitable for offline analysis, and it is unrealistic to assume the actual reservoir operation can be automatic based on any modeling results. This paper proposes the Real-time Optimization Model Enhanced by Data Assimilation (ROMEDA) method to integrate reservoir operators' justification and optimization modeling results for actual reservoir release decisions during the flooding season. Reservoir operators can choose when to adopt the modeling results according to their considerations which vary by person and by reservoir. ROMEDA also combines the models (optimization model and an unsteady flow routing model) with observed stages of long and on-channel reservoirs via data assimilation procedures, which update the reservoir storage (state) for the optimization and simulation models and also mitigate the effect of model and observation errors.

The advantage of ROMEDA method compared to the traditional single/multi-objective optimization methods (OPT-S and OPT-M) and historical operation records (HOR) is demonstrated through a case study with an on-channel reservoir. The results show that reservoir operators perform differently during a small and a large flood event in dealing with the tradeoff between flood control and hydropower generation. They behave aggressively in taking some risk in flooding for more hydropower generation during a small flood event, while conservatively during a large flood event by taking quicker and stronger measures for flood peak clipping. Such behavior difference is incorporated to ROMEDA, together with stage observations, for more realistic reservoir release decisions during a flood event. With the case study reservoir, the ROMEDA method, which integrates the advantages of both machine and



human, results in less flood risk than HOR and OPT-M and larger water use (hydropower)
benefit than HOR and OPT-S.

Possible future improvements to ROMEDA include a) the real-time reservoir operation

model with stochastic optimization considering inflow forecast uncertainty (with improved
forecast accuracy and lead time); b) the observation data (with enhanced accuracy); c) better
understanding of reservoir operators' real-world decision behaviors and choices. The
ROMEDA method can be easily applied to other real-time operation problems for a joint use
of optimization and data assimilation.

**Data availability**. All the code and data used in this study can be requested by contacting the
first author Jingwen Zhang at jingwenz@illinois.edu and/or the corresponding author Ximing
Cai at xmcai@illinois.edu.

**Author contributions**. JWZ and XMC developed the main ideas and implemented the
algorithms of the methods. JWZ and XHL collected the data used in the case study. JWZ,
XMC, XHL, PL and HW prepared the paper.

**Competing interests.** The authors declare that they have no conflict of interest.



## Acknowledgements


This study was supported by the Chinese Ministry of Science and Technology 973 Research
Program (2018YFC0407405), and the Excellent Young Scientist Foundation of the National
Natural Science Foundation of China (51422907, 51822908). The first author would like to
acknowledge the Chinese Scholarship Council (CSC) for supporting her PhD study at the
University of Illinois at Urbana-Champaign (UIUC).

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
