# Peer review of "Real-time reservoir flood control operation enhanced by data assimilation"

_Hydrology and Earth System Sciences, 2020_

## Referee Comment (RC1) · Anonymous Referee #1 · 20 Aug 2020

The article describes a decision support system for the real-time operation of water reservoirs. The system, named ROMEDA, integrates some methodologies that are well established in the water management area, namely optimization and data assimilation. While the topic is probably of interest for this community, I found the paper to be very weak.

The first important problem is the lack of novelty: the problem of integrating optimization algorithms in decision support systems has been tackled for decades—with the rationale, as rightfully pointed out by the authors, of aiding decision-makers, rather than controlling reservoirs in a fully automated fashion. There are indeed many tools that can tackle reservoir operation problems, such as HEC-ResSim (US Army Corps

of Engineers), MIKE HYDRO Basin (DHI), or FEWS (Deltares). Importantly, all of these tools integrate optimization algorithms with different kinds of hydrologic-hydraulic models. Some of them, such as FEWS, use Data Assimilation routines. Therefore, I think that ROMEDA does not represent a step forward in the domain of decision support systems.

Another problem is the way with which the manuscript is conceptualised. I think any reader would expect to see a demonstration of the decision support system, with emphasis on a comparison with the "human's mental model" vaguely mentioned by the authors in the Introduction (see Figure 1). Instead, the manuscript shifts its emphasis on the methods underpinning ROMEDA, which, as mentioned above, are not novel.

I also have a gripe about the experimental setup, which is not clear, transparent, and reproducible. If the goal is to carry out a comparison between human operators and ROMEDA, I would then expect to read a detailed explanation of the operators behaviour (or of the rules that they must follow), rather than the confusing description provided at the beginning of Section 4. Unfortunately, the quality of the presentation is not a problem limited to Section 4, but an issue spanning across the entire document.

My suggestion is to decline the manuscript.

---

## Referee Comment (RC2) · Anonymous Referee #2 · 5 Sep 2020

This manuscript presents a method called ROMEDA which integrates simulation, optimization and data assimilation to operate a reservoir in real-time. The main contribution is the development of a human-machine interactive method for real-time reservoir operation. Actually, especially for a complex operational problem, an operator uses a decision support system as a tool to find the possible optimal solutions and chooses a solution based on his/her experience for the actual implementation. Thus, it is not a novel concept in the field of the decision support systems. The specific comments are as follows:

Simulation model

As the Saint-Venant equations are used to simulate the stream flow along a long channel (658 km), it is required to explain how to consider the inflow into each cross-section

from a large catchment (5600 sq-km). Instead, inflow hydrographs at the reservoir are shown in Figure 7 and 8.

Reservoir routing or flood routing is usually used in the reservoir operation study. This study uses unsteady flow routing for a long and narrow reservoir. The reason to use unsteady flow routing is described in Line 308-311. As the reservoir releases are controlled for two objectives (one control point), the simulation model could be simplified by using a mass balance equation and the area-capacity curve.

Optimization model

As two objectives are considered, there may be more than a single optimal solution and how to choose a solution in PADDS should be explained.

ROMEDA

In general, an optimization-simulation approach is used to operate a reservoir or a reservoir system and the operators choose a solution based on their experiences when there exists a number of possible optimal solutions. ROMEDA highlights the effectiveness of the human-machine interactive method for real-time reservoir operation. It is demonstrated with a way that the operator accepts or rejects the model result according to the storage threshold. This demonstration is very sample and does not present a human-machine interaction practically.

Case study

It is very strange for me to see a bed-profile of a natural river behind a reservoir. The river bed levels are up and down in many places and the bed level is very close to -50 m between section 10 and 11 (Figure 6). Therefore, a brief explanation is required to understand the bed-profile of this river. The unit of the longitudinal distance (km) should be mentioned in Figure 6. In addition, the area-capacity curve should be provided to show how the water stores in on-channel reservoir.

I have no major issues with the results, discussions and conclusions. However, the

most important problem is that ROMEDA does not involve a novel formulation for real-time operation of a reservoir. It will be more interesting to the readers if the authors emphasize on how the decision-maker's experience or behavior can be effectively integrated into a decision-support system to solve a real-time control problem.

Therefore, I suggest that a major revision is required to improve the methodology and to investigate the new experiment for further consideration.

---

## Author Comment (AC1) · 19 Sep 2020

The article describes a decision support system for the real-time operation of water reservoirs. The system, named ROMEDA, integrates some methodologies that are well established in the water management area, namely optimization and data assimilation. While the topic is probably of interest for this community, I found the paper to be very weak.

Reply: Thanks for your comments.

The first important problem is the lack of novelty: the problem of integrating optimization algorithms in decision support systems has been tackled for decades—with the rationale, as rightfully pointed out by the authors, of aiding decision-makers, rather

than controlling reservoirs in a fully automated fashion. There are indeed many tools that can tackle reservoir operation problems, such as HEC-ResSim (US Army Corps of Engineers), MIKE HYDRO Basin (DHI), or FEWS (Deltares). Importantly, all of these tools integrate optimization algorithms with different kinds of hydrologic-hydraulic models. Some of them, such as FEWS, use Data Assimilation routines. Therefore, I think that ROMEDA does not represent a step forward in the domain of decision support systems.

Reply: First, please allow us to clarify the purpose of our work. We aim at a real-time human-machine interactive method for reservoir operation during a flooding event, using data assimilation of real-time observations to reduce the uncertainty from the simulation model. This method is new (according to our knowledge) by directly linking the reservoir operator with a traditional real-time reservoir operation model (integrated optimization and simulation). The computer model runs by rolling time windows. For one time window, it assimilates the observation of water levels and adopt the actual release the operator makes (which can be the same or different from the model recommended optimal value) at the end of the time period, and then moves to next and generates recommended release again. Meanwhile, the operator checks the recommended release from the model during each time window and decides to take it or do it differently based on their own justification. As you mentioned, HEC-ResSim (US Army Corps of Engineers), MIKE HYDRO Basin (DHI), or FEWS (Deltares) all tackle reservoir operation using optimization algorithms coupled with as hydrologic or hydraulic simulation, as well as data assimilation. However, all these decision support systems are not used in the way as ROMEDA by reservoir operators. They can provide decision support information, which can be used or not by the real reservoir operators, but they do not track the actual decisions made by the operators. In other words, ROMEDA proposes "online" interactions between model and user while such interactions with existing DSSs are usually "off-line." Besides, the data assimilation routine in FEWS (Deltares) handles predictive environmental disturbance, such as weather forecast uncertainty, similar to Model Predictive Control (MPC) (Camacho & Alba, 2013; Garcia et al., 1989; Macian-

Sorribes & Pulido-Velazquez, 2019), while the data assimilation in ROMEDA mitigates the uncertainty from the simulation model (e.g. a 1D hydrodynamic model solved by Preissmann) that takes into account actual releases made by reservoir operators. Thus regarding the novelty, we would argue that this study proposes an online (or real-time) human-machine interactive method for reservoir operation.

References:

Camacho, E. F., and Alba, C. B.: Model predictive control, Springer Science & Business Media, 2013.

Garcia, C. E., Prett, D. M., and Morari, M.: Model predictive control: theory and practice—a survey, Automatica, 25, 335-348, 1989.

Macian-Sorribes, H., and Pulido-Velazquez, M.: Inferring efficient operating rules in multireservoir water resource systems: A review, Wiley Interdisciplinary Reviews: Water, n/a, e1400, 10.1002/wat2.1400, 2019.

Another problem is the way with which the manuscript is conceptualised. I think any reader would expect to see a demonstration of the decision support system, with emphasis on a comparison with the "human's mental model" vaguely mentioned by the authors in the Introduction (see Figure 1). Instead, the manuscript shifts its emphasis on the methods underpinning ROMEDA, which, as mentioned above, are not novel.

Reply: Thanks for your comments. Indeed some studies (Hejazi & Cai., 2011; Hejazi et al., 2008; Castelletti et al., 2010) couple a "mentor model" (made via machine learning methods such as ANN) with a numerical simulation/optimization model to explore better reservoir operation plans. However, this is not what we want to do in this paper. As stated above, reservoir operators directly interact with the model (coupled simulation and optimization) and thus there is no need to use a computer-based mentor model to mimic the operators' behaviors. By the way, such models are usually limited in its effectiveness.

References:

Hejazi, Mohamad I., and Ximing Cai. "Building more realistic reservoir optimization models using data mining–A case study of Shelbyville Reservoir." Advances in water resources 34.6 (2011): 701-717.

Hejazi, Mohamad I., Ximing Cai, and Benjamin L. Ruddell. "The role of hydrologic information in reservoir operation–learning from historical releases." Advances in water resources 31.12 (2008): 1636-1650.

Castelletti, A., et al. "Tree‐based reinforcement learning for optimal water reservoir operation." Water Resources Research 46.9 (2010).

I also have a gripe about the experimental setup, which is not clear, transparent, and reproducible. If the goal is to carry out a comparison between human operators and ROMEDA, I would then expect to read a detailed explanation of the operators behaviour (or of the rules that they must follow), rather than the confusing description provided at the beginning of Section 4. Unfortunately, the quality of the presentation is not a problem limited to Section 4, but an issue spanning across the entire document.

Reply: As stated above, the construction of a human's mental model to mimic the reservoir operators' behaviors/experiences/considerations is not the purpose of this paper. However, we have to admit that in the demonstration of the ROMEDA method, we assume some simple (but reasonable) rules for the interaction between operators and the model, i.e., reservoir operators do not follow the model suggested reservoir releases but take some actions based on their own consideration and experiences, when the storage is below the maximum storage required for leaving space for coming storms. This is one of the possible ways of the operators and the model interact. As reservoir operators' considerations vary by person and by reservoir and involve multiple factors, such as policies and regulations, how to set more realistic rules for an operator to follow or not follow the modeled recommended releases is worthy of additional research, which is beyond the scope of this study, i.e., demonstrating that

the proposed ROMEDA works.

---

## Author Comment (AC2) · 19 Sep 2020

This manuscript presents a method called ROMEDA which integrates simulation, optimization and data assimilation to operate a reservoir in real-time. The main contribution is the development of a human-machine interactive method for real-time reservoir operation. Actually, especially for a complex operational problem, an operator uses a decision support system as a tool to find the possible optimal solutions and chooses a solution based on his/her experience for the actual implementation. Thus, it is not a novel concept in the field of the decision support systems. The specific comments are as follows:

Reply: Thanks for your comments. First, please allow us to clarify the purpose of our

[Figure]

work. We aim at a real-time human-machine interactive method for reservoir operation during a flooding event, using data assimilation of real-time observations to reduce the uncertainty from the simulation model. This method is new (according to our knowledge) by directly linking the reservoir operator with a traditional real-time reservoir operation model (integrated optimization and simulation). The computer model runs by rolling time windows. For one time window, it assimilates the observation of water levels and adopt the actual release the operator makes (which can be the same or different from the model recommended optimal value) at the end of the time period, and then moves to next and generates recommended release again. Meanwhile, the operator checks the recommended release from the model during each time window and decides to take it or do it differently based on their own justification. Thus regarding the novelty, we would argue that this study proposes an online (or real-time) human-machine interactive method for reservoir operation.

Simulation model

As the Saint-Venant equations are used to simulate the stream flow along a long channel (658 km), it is required to explain how to consider the inflow into each cross-section from a large catchment (5600 sq-km). Instead, inflow hydrographs at the reservoir are shown in Figure 7 and 8.

Reply: Thanks. The reservoir authority provides the inflow data for the on-channel reservoir, including the streamflow from an upstream section (section 1) and lateral flow (q in Eq 1) within one segment (i.e., at 489 km away from dam) These two sources are inputs to the Saint-Venant equations (Eq 1 and 2) simulating the on-channel reservoir. The inflow hydrographs in Figure 7 and 8 are the sum of two sources of inflow to reflect the magnitude and variation of the inflow. More explanation will be added in the manuscript.

Reservoir routing or flood routing is usually used in the reservoir operation study. This study uses unsteady flow routing for a long and narrow reservoir. The reason to use

unsteady flow routing is described in Line 308-311. As the reservoir releases are controlled for two objectives (one control point), the simulation model could be simplified by using a mass balance equation and the area-capacity curve.

Reply: It is not appropriate to treat the surface of the on-channel reservoir with a significant slope as a flat surface during the flooding season. Thus, it is not appropriate to simulate the reservoir flood routing by a static storage-stage relationship assuming a flat surface. Besides, we use the stage (water level)-capacity curve rather than the area-capacity curve in reservoir simulation. This is because the stage (water level)-capacity curve is usually determined under a flat surface condition, without considering the dynamic reservoir storage (i.e. the spatial variability of water level for the on-channel reservoir). Thus, the 1-D unsteady flow routing model is used to simulate the flood routing in the on-channel reservoir to calculate the dynamic reservoir storage.

Optimization model

As two objectives are considered, there may be more than a single optimal solution and how to choose a solution in PADDS should be explained.

Reply: Thanks. Flood control and hydropower generation both are considered as objectives in the multi-objective optimization model, and PADDS is the algorithm used to solve the multi-objective problem Since flood control is the primary objective during the flooding season, the solution in favor of flood control in the Pareto frontier is selected. More explanation will be added in the manuscript.

ROMEDA

In general, an optimization-simulation approach is used to operate a reservoir or a reservoir system and the operators choose a solution based on their experiences when there exists a number of possible optimal solutions. ROMEDA highlights the effectiveness of the human-machine interactive method for real-time reservoir operation. It is demonstrated with a way that the operator accepts or rejects the model result according to the storage threshold. This demonstration is very sample and does not present a human-machine interaction practically.

Reply: Thanks. Indeed some studies (Hejazi  Cai., 2011; Hejazi et al., 2008; Castelletti et al., 2010) couple a "mentor model" (describing reservoir operators' experiences made via machine learning methods such as ANN) with a numerical simulation/optimization model to explore better reservoir operation plans. However, this is not what we want to do in this paper. However, we have to admit that in the demonstration of the ROMEDA method, we assume some simple (but reasonable) rules for the interaction between operators and the model, i.e., reservoir operators do not follow the model suggested reservoir releases but take some actions based on their own consideration and experiences, when the storage is below the maximum storage required for leaving space for coming storms. This is one of the possible ways of the operators and the model interact. As reservoir operators' considerations vary by person and by reservoir and involve multiple factors, such as policies and regulations, how to set more realistic rules for an operator to follow or not follow the modeled recommended releases is worthy of additional research, which is beyond the scope of this study, i.e., demonstrating that the proposed ROMEDA works.

References:

Hejazi, Mohamad I., and Ximing Cai. "Building more realistic reservoir optimization models using data mining–A case study of Shelbyville Reservoir." Advances in water resources 34.6 (2011): 701-717.

Hejazi, Mohamad I., Ximing Cai, and Benjamin L. Ruddell. "The role of hydrologic information in reservoir operation–learning from historical releases." Advances in water resources 31.12 (2008): 1636-1650.

Castelletti, A., et al. "Tree‐based reinforcement learning for optimal water reservoir operation." Water Resources Research 46.9 (2010).

Case study

It is very strange for me to see a bed-profile of a natural river behind a reservoir. The river bed levels are up and down in many places and the bed level is very close to -50 m between section 10 and 11 (Figure 6). Therefore, a brief explanation is required to understand the bed-profile of this river. The unit of the longitudinal distance (km) should be mentioned in Figure 6. In addition, the area-capacity curve should be provided to show how the water stores in on-channel reservoir.

Reply: The case reservoir in this paper is an on-channel reservoir with a long bed-profile. A brief explanation will be added to understand the bed-profile of the on-channel reservoir. The longitudinal distance (km) is added in Figure 6. As explained above, we use the stage (water level)-capacity curve rather than the area-capacity curve in the reservoir simulation.

Figure 6 Longitudinal profile of the on-channel reservoir (see below)

I have no major issues with the results, discussions and conclusions. However, the most important problem is that ROMEDA does not involve a novel formulation for real-time operation of a reservoir. It will be more interesting to the readers if the authors emphasize on how the decision-maker's experience or behavior can be effectively integrated into a decision-support system to solve a real-time control problem.

Reply: Thanks. As stated above, the construction of a human's mental model to mimic the reservoir operators' behaviors/experiences/considerations is not the purpose of this paper. However, this is not what we want to do in this paper. As stated above, reservoir operators directly interact with the model (coupled simulation and optimization) and thus there is no need to use a computer-based mentor model to mimic the operators' behaviors. By the way, such models are usually limited in its effectiveness. ROMEDA is new (according to our knowledge) by directly linking the reservoir operator with a traditional real-time reservoir operation model (integrated optimization and simulation). In other words, ROMEDA proposes "online" interactions between model and user while

such interactions with existing DSSs are usually "off-line."

Therefore, I suggest that a major revision is required to improve the methodology and to investigate the new experiment for further consideration.

Reply: Thanks for your valuable comments. We hope our reply can clarify the confusing part of the manuscript.

[Figure]

[Figure]

**Fig. 1.**

---

## Referee Comment (RC3) · Anonymous Referee #3 · 21 Sep 2020

In this paper, the authors propose a human-machine interactive method, namely Real-time Optimization Model Enhanced by Data Assimilation (ROMEDA) for reservoirs that have complex storage and stage relations (e.g. long and narrow reservoirs). The ROMEDA is essentially a decision support system for reservoir simulation and optimization. Authors conduct case studies to show that for both small and large flood events, ROMEDA shows better performance on flood risk mitigation and water use (hydropower) benefit than the case with historical operation records (HOR) or optimization with single/multi-objective. An on-channel reservoir for flood control from China is selected to test the proposed ROMEDA method. Results compare the proposed ROMEDA against three additional scenarios OPT-S, OPT-M, and historical release decisions. The OPT-S is the optimal decision obtained from the Dynamically Dimensioned

[Figure]

Search algorithm (DDS) (Tolson and Shoemaker, 2007) for single-objective optimization. The OPT-M is the optimal decision obtained from the Pareto archived dynamically dimensioned search algorithm (PADDS) (Jahanpour et al., 2018) for multiple objective optimizations.

First off, the reviewer does not agree with the authors' statement in the paper that "ROMEDA is one of the first attempts of a human-machine interactive method for online use of an optimization model for real-time reservoir operation based on integrated modeling, observation, and operators' choice." Two reasons:

1. Human-machine interactive method does not mean users simply play with the computer program and software. From computer science, it is a two-way interaction mechanism that machine also learns and improves the computation logic via human's inputs. The decision support system in water resources, specifically, reservoir operation falls into the one-way category, and it is not a unique feature of the proposed ROMEDA framework. Many existing software and decision support systems, for instance, OASIS, Delft-FEWS, Riverwares, HEC-RES, also include similar features that dynamic updating and re-calculation are parts of the software/program. The authors claimed "human-machine interative method" (Figure 1) as well as throughout the manuscript, is essentially only human define conditions/states for computers to generate outputs. Therefore, the novelty claimed by authors as "human-machine interactive method" for the ROMEDA is not convincing for me.

2. The merging of observation data, optimization algorithms, and operators' choice for quick and effective decision making is the goal of any decision support system, and it is not unique to the proposed ROMEDA framework. Refer to the UNESCO book "Water Resources Management and Planning" by Daniel P. Loucks and Eelco van Beek, any decision support system shall include the elements to inject data, the element to conduct simulation and optimization, the elements for presenting GIS information, and the elements to visualize the results for decision making. The integration of all those elements makes a decision support system. From this perspective, the ROMEDA system

does not provide any new features or different elements for a decision support system. In other words, what are the essential features, differences, and advancements of the proposed ROMEDA system as compared to the existing decision making support systems in the field of reservoir operation? This point is also raised by other referees.

Secondly, the reviewer thinks the experimental design does not illustrate the claimed advantage of the ROMEDA framework. Two reasons are listed below:

3. Authors claimed that "demonstrate that an unsteady flow routing simulation model is needed for reservoirs that are a long and narrow channel, for which it is not accurate enough to use a static storage-stage relationship to simulate the reservoir storage; while it is also impossible to measure the storage directly because the reservoir surface is not flat." Reviewer is wondering how significant are the uses of data assimilation technique in enhancing the accuracy of the unsteady flow routing model? In the presented case, the reviewer did not see any proof that the unsteady flow routing could be more accurate than the case using a static storage-stage relationship. The latter is a commonly accepted approach in many operational sectors as it is simple and accurate enough to do reservoir planning for hydropower, water supplies, flood control objectives, etc. If the authors could not demonstrate the significant amount of errors of this assumption, the technical foundation for the motivation of this study is questionable.

4. In line 206-210, authors confirmed that "ROMEDA is similar to Model Predictive Control (MPC) (Garcia et al., 1989; Camacho and Alba, 2013; Macian-Sorribes and Pulido-Velazquez, 2019) and other real time control approaches, such as on-line adaptive control (Soncini-Sessa et al., 2007), open-loop and closed- loop control (Soncini-Sessa et al., 2007; Gerdts, 2012) with respect to more effective use of computer-based models and observed data." In fact, the MPC is also a commonly accepted approach in many decision support system to account for forecast-informed operation. The reviewer is glad authors mentioned this, but was expecting the later experiments to compare the advances or improvements of the proposed ROMEDA framework v.s. a decision making support system with MPC. However, this did not happen in later experiments. The

only comparison were made with standard dynamic optimization schemes, Dynamically Dimensioned Search algorithm (DDS) and Pareto archived dynamically dimensioned search algorithm (PADDS) (Jahanpour et al., 2018). Then, the question still remains, why the proposed ROMEDA framework is any better than the MPC or similar techniques used for real time control of reservoir and hydraulic systems?

Last, about the experiment's results and the limitations of the ROMEDA system. The reviewer does not have problems understanding the presented results regarding the flood risks and hydropower generation improvements in the result section. However, the reviewer was not fully convinced that the presented experiments have considered the real challenges of real-time reservoir operation. Three reasons:

5. The presented results demonstrate the ROMEDA system is much better on flood risks and hydropower generation than historical release decisions. This is not surprising when using historical inflow as perfect forecasts (also see Line 562-564). In reality, reservoir operators do not know for sure how much inflows will be coming to the reservoir and channel systems until it happens. This essentially will cause the historical decision becomes more conservative than the optimized results. So, in theory, the optimized results could easily beat historical decisions when lifting or lowering the storage level correspondingly and creates some aggressive moves towards global optimal solutions when reanalyzing the historical inflow as inputs. The reviewer is especially interested in how ROMEDA treats the forecasts with uncertainties. However, the manuscript does not specifically test it out, say if perturb the inflow forecast by some random errors and see whether the conclusion could be changed.

6. The reservoir system being investigated in this manuscript may not represent other systems. The reviewer is wondering about the specific hydraulic constraints and the real operation rules guiding the release decisions. For example, whether there are soft constraints in the presented system, and besides flood control and hydropower, are there other objectives or constraints this system is designed for? I think this is also reflected in other referees' comments. How does the ROMEDA deal with environmental

constraints?

7. Last, the reviewer is interested in the runtime of the proposed ROMEDA framework. Could the authors demonstrate a few cases to show the runtime? Say, for one-day simulation at hourly steps? And a 30 day simulation at hourly steps? I think the presented results are less than a month, maybe authors could include some run time statistics of all the employed models as comparison.

———————————————————

---

## Author Comment (AC3) · 22 Sep 2020

In this paper, the authors propose a human-machine interactive method, namely Realtime Optimization Model Enhanced by Data Assimilation (ROMEDA) for reservoirs that have complex storage and stage relations (e.g. long and narrow reservoirs). The ROMEDA is essentially a decision support system for reservoir simulation and optimization. Authors conduct case studies to show that for both small and large flood events, ROMEDA shows better performance on flood risk mitigation and water use (hydropower) benefit than the case with historical operation records (HOR) or optimization with single/multi-objective. An on-channel reservoir for flood control from China is selected to test the proposed ROMEDA method. Results compare the proposed ROMEDA against three additional scenarios OPT-S, OPT-M, and historical release decisions. The OPT-S is the optimal decision obtained from the Dynamically Dimensioned Search algorithm (DDS) (Tolson and Shoemaker, 2007) for single-objective optimization. The OPT-M is the optimal decision obtained from the Pareto archived dynamically dimensioned search algorithm (PADDS) (Jahanpour et al., 2018) for multiple objective optimizations.

Reply: Thanks for your comments.

First off, the reviewer does not agree with the authors' statement in the paper that "ROMEDA is one of the first attempts of a human-machine interactive method for online use of an optimization model for real-time reservoir operation based on integrated modeling, observation, and operators' choice." Two reasons:

1. Human-machine interactive method does not mean users simply play with the computer program and software. From computer science, it is a two-way interaction mechanism that machine also learns and improves the computation logic via human's inputs. The decision support system in water resources, specifically, reservoir operation falls into the one-way category, and it is not a unique feature of the proposed ROMEDA framework. Many existing software and decision support systems, for instance, OASIS, Delft-FEWS, Riverwares, HEC-RES, also include similar features that dynamic updating and re-calculation are parts of the software/program. The authors claimed "human-machine interative method" (Figure 1) as well as throughout the manuscript, is essentially only human define conditions/states for computers to generate outputs. Therefore, the novelty claimed by authors as "human-machine interactive method" for the ROMEDA is not convincing for me.

Reply: First, please allow us to clarify the purpose of our work. We aim at a real-time human-machine interactive method for reservoir operation during a flooding event, using data assimilation of real-time observations to reduce the uncertainty from the simulation model. This method is new (according to our knowledge) by directly linking the reservoir operator with a traditional real-time reservoir operation model (integrated

optimization and simulation). The computer model runs by rolling time windows during the flooding control season. For one time window, it assimilates the observation of water levels and adopt the actual release the operator makes (which can be the same or different from the model recommended optimal value) at the end of the time period, and then moves to next and generates recommended release again. Meanwhile, the operator checks the recommended release from the model during a time window and decides to take it or do it differently based on their own justification. As you mentioned, Delft-FEWS, Riverwares, or HEC-RES all tackle reservoir operation using optimization algorithms coupled with as hydrologic or hydraulic simulation. However, all these decision support systems are not used in the way as ROMEDA by reservoir operators. They can provide decision support information, which can be used or not by the real reservoir operators, but they do not track the actual decisions made by the operators. The dynamic updating and re-calculation of these decision-support systems mainly deal with the forecast uncertainty, rather than the actual decisions made by the operators. In other words, ROMEDA proposes "online" interactions between model and user while such interactions with existing DSSs are usually "off-line."

Besides, the data assimilation routine in FEWS (Deltares) handles predictive environmental disturbance, such as weather forecast uncertainty, similar to Model Predictive Control (MPC) (Camacho & Alba, 2013; Garcia et al., 1989; Macian-Sorribes & Pulido-Velazquez, 2019), while the data assimilation in ROMEDA mitigates the uncertainty from the simulation model (e.g. a 1D hydrodynamic model solved by Preissmann) that takes into account actual releases made by reservoir operators.

And the expert system in OASIS is built based on artificial intelligence for groundwater contaminant modeling, rather than reservoir operation. The construction of the expert systems ("mentor model") is not what we want to do in this paper. As stated above, reservoir operators directly interact with the model (coupled simulation and optimization) and thus there is no need to use a computer-based mentor model to mimic the operators' behaviors. By the way, such models are usually limited in its effectiveness.

For your statement "From computer science, it is a two-way interaction mechanism that machine also learns and improves the computation logic via human's inputs. The decision support system in water resources, specifically, reservoir operation falls into the one-way category, and it is not a unique feature of the proposed ROMEDA framework." We totally agree that "The decision support system in water resources, specifically, reservoir operation falls into the one-way category", but ROMEDA indeed implements the two-way interaction – operators check the model results from time to time and take or not take the model recommended release decision; meanwhile the model adopts the actual decisions made by the operator via the data assimilation approach. Such interactions move forward period by period through the flooding control season.

Thus regarding the novelty, we would argue that this study proposes an online (or real-time) human-machine interactive method for reservoir operation. However, we thank the review to bring up the confusion and we will clarify the processes of ROMEDA and show it novelty clearly.

References:

Camacho, E. F., and Alba, C. B.: Model predictive control, Springer Science & Business Media, 2013. Garcia, C. E., Prett, D. M., and Morari, M.: Model predictive control: theory and practice—a survey, Automatica, 25, 335-348, 1989. Macian-Sorribes, H., and Pulido-Velazquez, M.: Inferring efficient operating rules in multireservoir water resource systems: A review, Wiley Interdisciplinary Reviews: Water, n/a, e1400, 10.1002/wat2.1400, 2019.

2. The merging of observation data, optimization algorithms, and operators' choice for quick and effective decision making is the goal of any decision support system, and it is not unique to the proposed ROMEDA framework. Refer to the UNESCO book "Water Resources Management and Planning" by Daniel P. Loucks and Eelco van Beek, any decision support system shall include the elements to inject data, the element to conduct simulation and optimization, the elements for presenting GIS information, and the

elements to visualize the results for decision making. The integration of all those elements makes a decision support system. From this perspective, the ROMEDA system does not provide any new features or different elements for a decision support system. In other words, what are the essential features, differences, and advancements of the proposed ROMEDA system as compared to the existing decision making support systems in the field of reservoir operation? This point is also raised by other referees.

Reply: As stated above, the main difference between ROMEDA and existing decision-support systems of reservoir operation (such as Delft-FEWS, Riverwares, and HEC-RES) is the online incorporation of the actual decisions made by the operators.

Secondly, the reviewer thinks the experimental design does not illustrate the claimed advantage of the ROMEDA framework. Two reasons are listed below:

3. Authors claimed that "demonstrate that an unsteady flow routing simulation model is needed for reservoirs that are a long and narrow channel, for which it is not accurate enough to use a static storage-stage relationship to simulate the reservoir storage; while it is also impossible to measure the storage directly because the reservoir surface is not flat." Reviewer is wondering how significant are the uses of data assimilation technique in enhancing the accuracy of the unsteady flow routing model? In the presented case, the reviewer did not see any proof that the unsteady flow routing could be more accurate than the case using a static storage-stage relationship. The latter is a commonly accepted approach in many operational sectors as it is simple and accurate enough to do reservoir planning for hydropower, water supplies, flood control objectives, etc. If the authors could not demonstrate the significant amount of errors of this assumption, the technical foundation for the motivation of this study is questionable.

Reply: The significance of data assimilation for the accuracy of the unsteady flow routing model is demonstrated in Section 5.1 (Figure 11).

It is not appropriate to treat the surface of the on-channel reservoir with a significant slope as a flat surface during the flooding season. Thus, it is not appropriate to simulate

the reservoir flood routing by a static storage-stage relationship assuming a flat surface. This is because the stage (water level)-capacity curve is usually determined under a flat surface condition, without considering the dynamic reservoir storage (i.e. the spatial variability of water level for the on-channel reservoir). Conventional studies for hydropower generation and water supply usually assume flat surface, while the surface for the on-channel reservoir during flood season usually cannot be assumed as flat. Thus, the 1-D unsteady flow routing model is used to simulate the flood routing in the on-channel reservoir to calculate the dynamic reservoir storage.

4. In line 206-210, authors confirmed that "ROMEDA is similar to Model Predictive Control (MPC) (Garcia et al., 1989; Camacho and Alba, 2013; Macian-Sorribes and PulidoVelazquez, 2019) and other real time control approaches, such as on-line adaptive control (Soncini-Sessa et al., 2007), open-loop and closed- loop control (Soncini-Sessa et al., 2007; Gerdts, 2012) with respect to more effective use of computer-based models and observed data." In fact, the MPC is also a commonly accepted approach in many decision support system to account for forecast-informed operation. The reviewer is glad authors mentioned this, but was expecting the later experiments to compare the advances or improvements of the proposed ROMEDA framework v.s. a decision making support system with MPC. However, this did not happen in later experiments. The only comparison were made with standard dynamic optimization schemes, Dynamically Dimensioned Search algorithm (DDS) and Pareto archived dynamically dimensioned search algorithm (PADDS) (Jahanpour et al., 2018). Then, the question still remains, why the proposed ROMEDA framework is any better than the MPC or similar techniques used for real time control of reservoir and hydraulic systems?

Reply: As you mentioned, MPC is proposed to deal with the uncertainty of weather forecast, while ROMEDA is proposed as an online incorporation of the actual decisions made by the operators. MPC can be incorporated into ROMEDA to investigate the impact of forecast uncertainty. However, this is not what we want to do in this paper. In summary, ROMEDA is different from other real-time reservoir control systems because

of its online incorporation of the reservoir operators' experience (actual decisions).

Last, about the experiment's results and the limitations of the ROMEDA system. The reviewer does not have problems understanding the presented results regarding the flood risks and hydropower generation improvements in the result section. However, the reviewer was not fully convinced that the presented experiments have considered the real challenges of real-time reservoir operation. Three reasons:

5. The presented results demonstrate the ROMEDA system is much better on flood risks and hydropower generation than historical release decisions. This is not surprising when using historical inflow as perfect forecasts (also see Line 562-564). In reality, reservoir operators do not know for sure how much inflows will be coming to the reservoir and channel systems until it happens. This essentially will cause the historical decision becomes more conservative than the optimized results. So, in theory, the optimized results could easily beat historical decisions when lifting or lowering the storage level correspondingly and creates some aggressive moves towards global optimal solutions when reanalyzing the historical inflow as inputs. The reviewer is especially interested in how ROMEDA treats the forecasts with uncertainties. However, the manuscript does not specifically test it out, say if perturb the inflow forecast by some random errors and see whether the conclusion could be changed.

Reply: This paper aims to demonstrate the superiority of the combination of reservoir operators' experience and traditional optimization models (i.e. human-machine interactive method). Based on the results, ROMEDA system performs better than historical operation (only with operators' experience) and OPT (only with optimization models without incorporating human interactions). As stated above, MPC can be incorporated into ROMEDA to investigate the impact of inflow uncertainty. However, this is not what we want to do in this paper. We have discussed the forecast uncertainty in section 5.3. For the application of ROMEDA in this manuscript, the impact of inflow forecast can be ignored for two reasons. Firstly, the uncertainty of inflow forecast with a 72-hour forecast horizon is relatively low due to the advanced forecast technologies. Secondly,

the results of ROMEDA still performs better than the traditional optimization models with perfect inflow forecast.

6. The reservoir system being investigated in this manuscript may not represent other systems. The reviewer is wondering about the specific hydraulic constraints and the real operation rules guiding the release decisions. For example, whether there are soft constraints in the presented system, and besides flood control and hydropower, are there other objectives or constraints this system is designed for? I think this is also reflected in other referees' comments. How does the ROMEDA deal with environmental constraints?

Reply: That is correct. Actually, actual reservoir operation is impact by multiple factors, such as policies and regulations, environmental constraints, and various objectives, which cannot be fully considered in the traditional optimization models. Thus, ROMEDA aims to incorporate the reservoir operators' experience (almost cover all factors) into the traditional optimization model (cover limited factors) for better performance than historical operation and traditional optimization models.

7. Last, the reviewer is interested in the runtime of the proposed ROMEDA framework. Could the authors demonstrate a few cases to show the runtime? Say, for one-day simulation at hourly steps? And a 30 day simulation at hourly steps? I think the presented results are less than a month, maybe authors could include some run time statistics of all the employed models as comparison.

Reply: The simulation time of ROMEDA is more than the traditional optimization models, as data assimilation is applied to reducing the model uncertainty. And the simulation time of ROMEDA mainly depends on the ensemble size of EnKF. Two flood events (small flood event and large flood event) are tested using ROMEDA. The duration of small flood event lasts for 403 hours, and that of large flood event lasts for 635 hours. The statistics of run time can be added if needed.

304, 2020.